# Effects of Shading Nets Color on the Internal Environmental Conditions, Light Spectral Distribution, and Strawberry Growth and Yield in Greenhouses

**DOI:** 10.3390/plants13162318

**Published:** 2024-08-20

**Authors:** Ibrahim M. Alhelal, Ammar A. Albadawi, Abdullah A. Alsadon, Mekhled M. Alenazi, Abdullah A. Ibrahim, Mohamed Shady, Abdulhakim A. Al-Dubai

**Affiliations:** 1Department of Agricultural Engineering, College of Food and Agriculture Sciences, King Saud University, P.O. Box 2460, Riyadh 11451, Saudi Arabia; imhelal@ksu.edu.sa (I.M.A.); mshady@ksu.edu.sa (M.S.); 2Department of Plant Production, College of Food and Agriculture Sciences, King Saud University, P.O. Box 2460, Riyadh 11451, Saudi Arabia; amekhled@ksu.edu.sa (M.M.A.); adrahim@ksu.edu.sa (A.A.I.); hakimaldobai@gmail.com (A.A.A.-D.)

**Keywords:** color shading, light spectrum, photosynthesis, transpiration

## Abstract

Greenhouses are used to create the appropriate environment for plant growth. Controlling the level of lighting using shading nets is one of the most commonly used methods for making suitable environmental modifications in greenhouses. The objective of this study was to examine the impact of three colored shading nets (green, black, and beige at shading rates of 50%) on inside air temperature, relative humidity, and spectral distribution of light in a greenhouse, as well as their effect on the growth and yield of strawberry plants. Data were collected during winter (December and January) and spring (March and April) months from shaded and unshaded blocks. The green net had the highest transmittance to solar radiation (τ_SR_) during the two periods (38% and 35%, respectively) and the highest transmittance to photosynthetically active radiation (τ_PAR_) of 34% during spring months, while the beige net had the highest τ_PAR_ of 27% during winter months. The black net had the smallest τ_PAR_ values during the two periods (22% and 29%, respectively). The lowest total light levels per season for solar radiation (SR) and photosynthetically active radiation (PAR) (746.8 and 293.7 MJ·m^−2^, respectively) were obtained under the black net, compared with (906.7 and 320.8 MJ·m^−2^, respectively) for the beige net, and (969.6 and 337.2 MJ·m^−2^, respectively) for the green net. The ratio of PAR to SR (PAR:SR) was 41% and 44% outside and inside the greenhouse for the control (without shade), respectively. The black net had the highest ratio of PAR:SR (39%) among the treatment nets. The green net transmitted more light in the blue–green region (400 to 570 nm) and transmitted the highest photon flux at 480 nm, while the beige net increased the infrared radiation flux from 730 nm and above and transmitted the highest photon flux at 604 nm. The study found that the green net increased the ratio of blue to red light (B/R), while the beige and green nets reduced the red to far-red light (R/FR) ratio. The photosynthetic rate, conductance to water, and transpiration were significantly higher for strawberries grown under the beige net. These results indicate that the beige net positively influenced leaf and stem characteristics, leading to improved strawberry yields. The best yields of strawberries were obtained under the beige net and the control group (no shade), surpassing the yields achieved under the black net by 26.3% and 21.4%, respectively.

## 1. Introduction

Greenhouses are commonly used for crop production in the Arabian Peninsula, but the hot and sunny summer conditions present a challenge for growers due to the large amount of solar radiation that enters the greenhouse. Shading nets made of knitted plastic threads with air pores are commonly used to modify the environment and reduce heat accumulation. These nets also have the ability to cut out and diffuse solar radiation, as well as improve cooling efficiency and ventilation. Also, shading nets can contribute to optimizing crop productivity and positively affecting its quality in adverse climatic conditions, especially in hot and sunny regions.

Understanding the optical characteristics of greenhouse coverings, along with crop requirements, is important to achieve successful crop growth. The transmission of solar radiation is a crucial factor for crop production and is used for characterizing different types of shading nets. Measuring the actual transmissivity of covering materials is essential for efficient greenhouse operation and for determining the suitable shading net for each crop. Growers should have a good knowledge of these properties.

The use of various combinations of greenhouses covering nets has become of interest to growers in maintaining ideal conditions for crop production. Several studies were carried out to characterize the environmental conditions under plastic coverings. Currently, many types of plastic coverings are available with different optical characteristics. Using shading screens with special optical properties can enable producers to modify the composition of transmitted solar radiation and improve crop production [1]. According to Robledo de P and Marín [2], the color and transparency of covering materials affect radiation absorption, reflection, and transmission.

Differences between plastic screens can be based on solidity, level of shading, chemical composition, and coloration. Zhang, et al. [3] stated that shading with black, blue, or red shade nets had overall positive effects on tea plant growth index, net photosynthetic rate, and stomatal conductance compared to the no-shade control in the southeastern United States. Furthermore, it lowered air and leaf surface temperatures during the summer months and provided protection against cold temperatures in winter. The use of shade nets in lettuce production during the summer season in Serbian climate conditions had positive effects on lettuce growth and morphology, with varying impacts observed among different color shade nets, with the pearl shade net exhibiting the highest levels of phenols, flavonoids, and antioxidant properties [4]. They also concluded that shade nets rated at 50% significantly reduced solar irradiation from 890 W·m^−2^ to a range of 400–560 W·m^−2^ and decreased photosynthetically active radiation (PAR) from 2020 μmol·m^−2^·s^−1^ to less than 1000 μmol·m^−2^·s^−1^ when compared to open field conditions. In a study conducted by Díaz-Pérez and John [5], it was observed that the use of shade nets had a positive impact on the growth and leaf gas exchange of bell pepper plants. The primary benefit of shade nets was attributed to their ability to reduce leaf and root zone temperatures. Interestingly, the effect of shade net color on plant growth variables such as plant fresh weight and stem diameter was found to be inconsistent or relatively minor.

The use of colored shading nets in protected cultivation can enhance specific plant morphological and physiological traits that improve yield and quality traits, which lead to economic advantages [6]. Shade nets can modify the light spectrum and create ideal light conditions for plant growth [7]. They are employed to reduce light intensity and scatter incoming light by approximately 50% [8]. However, the effects of shade nets on plants can vary depending on the crops and environmental conditions in which they are cultivated. Stamps and Chandler [9] conducted a study that demonstrated the preference of different plant species for specific shade net colors. They found that the cast iron plant (*Aspidistra elatior* L.) showed optimal growth under black shade, while the Japanese pittosporum (*Pittosporum tobira* T.) performed better under red shade.

The study of Singh et al. [10] demonstrated that the color of shade nets affected ornamental plant species (celosia, begonia, gerbera, and fountain grass) growth and chlorophyll concentration, with aluminite shade nets showing positive effects on shoot dry weight in begonia and celosia. However, flower number was not influenced by the shade net color in the tested plant species. In a study conducted by Torres-Hernández et al. [11], it was observed that lisianthus (*Eustoma grandiflorum*) exhibited elongated stems with increased diameter when exposed to red shade. On the other hand, the minimum leaf area was achieved when lisianthus plants were subjected to blue shade. Colored screens provide an alternative to using plant regulators, as they can modify the solar radiation spectrum and provide physical protection for the plants [12]. However, the choice of colored covering materials is often made based on empirical or economic criteria rather than technical or scientific principles [13]. This is partly due to the lack of technical data on the optical properties of different plastics and shading nets and their effects on the microclimate. Castellano et al. [14] found that producers often may not select an ideal type of covering for a specific application and that further research is needed to characterize different types of screens for specific objectives. Developing an international standard method to identify the characteristics of these materials would be useful for producers.

Strawberry (*Fragaria x ananassa*. cv Fortuna), a hybrid of two highly variable octoploid species, is a small fruit crop that has adapted to different environmental conditions [15]. In Saudi Arabia, strawberries are grown in greenhouses where sunlight and available water are crucial for producing high-quality fruit [16,17,18]. Temperature is one of the most important factors affecting strawberry plant nutrient uptake [19], and is associated with flower bud induction, runner development, leaf variegation, fruit flavors, and membrane phospholipids [15,20,21,22,23].

Accordingly, it is important to investigate the influence of shading net color on the internal environment of greenhouses cultivated with strawberries under arid conditions, aiming to better assess the effect of optical properties of the shading screens on strawberry growth and yield.

## 2. Materials and Methods

### 2.1. The Greenhouse

The experiment was conducted in a greenhouse located at the protected cultivation research unit (24°43′33.21″ N, 46°36′50.41″ E), King Saud University. The greenhouse is covered by an 8 mm double-layer polycarbonate sheet and is oriented from the N-S direction. The dimensions of the greenhouse are 20 m in length, 8 m in width, 4.5 m in gutter height, and 2 m in gable height.

The greenhouse was equipped with a fan pad cooling system. The cooling pads were cross-fluted cellulose pads (CELdek^®^, Munters Corp., Chiusavecchia, Italy) measuring 7 m in width and 2 m in height. They were positioned at a height of 0.5 m above the greenhouse floor. To ensure proper ventilation, the greenhouse utilized two exhaust fans, each with a flow rate of 667 m^3^/min (equivalent to 1.5 air exchanges per minute). The fans were regulated by a control unit that maintained an air temperature set point between 25 °C and 27 °C. Additionally, the greenhouse has a drip irrigation system, with one 4 L dripper serving two plants that were transplanted in each pot. Fertilizers were added to the irrigation water through fertigation techniques.

### 2.2. Plastic Shading Nets

Three plastic shading nets that have a 50% shading rate (green, black, and beige) were used inside the greenhouse (Figure 1). Each shading net covered three blocks (replication) with dimensions of 5 m × 1.5 m × 3.5 m. These nets were locally produced by the Saudi Yarn and Knitted Technology Factory, Jeddah, Kingdom of Saudi Arabia (SYNTECH-ISO 9001).

### 2.3. Study Layout

The greenhouse was divided into 12 blocks, and the experiment was conducted in a randomized block design with 9 treatments and 3 replications (i.e., 3 shaded blocks, 3 non-shaded blocks with 3 replicates) (Figure 2). The three blocks were not covered with shade and were used as controls.

### 2.4. The Crop

The study used chilled strawberry (Firgo) seedlings that were obtained from the strawberry cultivation center at Ain-Shams University Cairo, Egypt. The cultivar is a short-day strawberry variety. The soil was a mixture of sand, clay, peat moss, vermiculite, perlite, and potting soil, with ratios of 50%, 25%, 6.25%, 6.25%, 6.25%, and 6.25%, respectively. The soil mixture was well-mixed and used for cultivation in 0.25 m pots, each containing two plants. The strawberry growing season was divided into three stages: (1) the vegetative growth stage, which lasts for 60 days from transplant to flowering; (2) the flowering and fruit setting stage, which lasts for 30 days; and (3) the fruit development and harvest (fruiting) stage, which lasts for 90 days. In order to enhance the vegetative growth, stolons and flowers were removed during the first stage. The fertilization and irrigation systems were uniform for all plants. Experiments were carried out during the period from 1 November 2022 to 30 May 2023, which was divided into two periods: period 1, representing winter months (December and January), and period 2, which represents spring months (March and April).

### 2.5. Measurements and Instrumentation

(i)The greenhouse outside and inside air temperature and relative humidity were measured using a combined sensor data logger (OM-EL-USB-2-LCD, Omega Inc., Norwalk, CT, USA). The sensors were programmed to take the experimental measurements every 5 min (Figure 3).(ii)The global solar radiation flux (SR) and the photosynthetically active radiation (PAR) were measured using the Kipp & Zonen CMP3 Pyranometer and LI-COR LI-190R Quantum Sensors, respectively. The SRg and PAR were measured under each of the shading blocks, unshaded blocks, and outside. The sensors were programmed to scan the experimental measurements every minute, and the average was recorded every five minutes. Data were collected using a LI-COR 1400 data logger (Figure 3).(iii)The spectral distribution of light was measured every 15 days using the Stellar Net BLACK-Comet Spectrometer at noon throughout the strawberry growth period.(iv)The photosynthesis rate, transpiration rate, and CO_2_ concentration were measured using a portable photosynthesis measuring system (LI-6400; Li-COR, Inc., Lincoln, NE, USA) at noon during the mid-stage of strawberry growth.

### 2.6. Calculation of Radiation Energy

(i)Global solar radiation flux (SR)

Total energy per day, total energy during all of the season, and transmittance through the net of (SR) were calculated by Equations (1), (2), and (3), respectively [24]:(1)$${SR}_{t,d}=\int_{t_{1}}^{t_{2}}SR\;dt$$
(2)$${SR}_{t}=\sum_{1}^{180}{(SR}_{t,d})\times 10^{-6}$$
(3)$$\tau_{net,SR}=\frac{{SR}_{t,in}}{{SR}_{t,out}}\times 100$$
where SR_t,d_ is the total energy of global solar radiation per day (J·m^−2^·day^−1^); t_1_ is the time of sunrise; t_2_ is the time of sunset; SR_t_ is the total energy of global solar radiation during all of the season (MJ·m^−2^·season^−1^) and τ_net,SR_ is the transmittance of global SR through the net. The subscripts “in” is the inside of the greenhouse under different blocks, and “out” is the outside of the greenhouse.

(ii)Photosynthetically active radiation (PAR)

Total photosynthetic photon flux density per day, total photosynthetic photon flux density during all of the season, and transmittance through the net of photosynthetically active radiation (PAR) were estimated by Equations (4), (5), and (6), respectively:(4)$${PAR}_{d}=\int_{t_{1}}^{t_{2}}PAR\;dt$$
(5)$${PAR}_{t}=\sum_{1}^{180}{(PAR}_{d})\times 10^{-6}$$
(6)$$\tau_{net,PAR}=\frac{{PAR}_{t,in}}{{PAR}_{t,out}}\times 100$$
where PAR_d_ is the total photosynthetic photon flux density per day (µmol·m^−2^·day^−1^); PAR_t_ is the total photosynthetic photon flux density of PAR during all of the season (mol·m^−2^·season^−1^); and τ_net,PAR_ is the transmittance of PAR through the net (note: the season = 180 days, and µmol·m^−2^·s^−1^ = 4.57 W·m^−2^ at 400–700 nm from solar radiation [25]).

(iii)Light spectral distribution calculations

The photon flux density of global solar radiation used light spectral distribution was defined by the equation [26]:(7)$${SR}_{g}=\int_{200}^{2500}G_{\lambda }\;d\lambda$$
where G is the spectral irradiance of the solar spectrum (µmol·m^−2^·nm^−1^), and λ is the spectral values. Photosynthetic photon flux density of (PAR) was defined by:(8)$${SR}_{PAR}=\int_{400}^{700}G_{\lambda }\;d\lambda$$

Photon flux density or photosynthetic photon flux density of blue, red, and far-red light spectrum were defined by Equations (9), (10), and (11), respectively:(9)$${SR}_{blue}=\int_{400}^{500}G_{\lambda }\;d\lambda$$
(10)$${SR}_{red}=\int_{600}^{700}G_{\lambda }\;d\lambda$$
(11)$${SR}_{far-red}=\int_{700}^{800}G_{\lambda }\;d\lambda$$

The blue light to red light ratio (BL/RD) and ratio of red light to far-red light (RD/FR) are defined by Equations (12) and (13), respectively:(12)BLRD=∫400500Gλ dλ∫600700Gλ dλ
(13)RDFR=∫600700Gλ dλ∫700800Gλ dλ

### 2.7. Growth and Yield Measurements

Some growth parameters were recorded and averaged for three plants from each block at the flowering stage (75 days after transplanting). Plant height (cm) was measured from the soil level of the pots to the top of the plant. The leaf area was measured using a Portable Area Meter (LI-COR model 3000 A), and the total number of crowns was also counted. The fresh weight of the plant shoot for the leaves and stems was measured. Additionally, the dry weight of the shoot was determined after drying the fresh sample of leaves and stem in a forced air oven at 70 °C for 48–72 h until a constant weight was achieved. At the fruiting stage, the total number of harvested fruits per plant was recorded, along with the fruit yield per plant in kg. The average fruit weight (gm) was calculated by dividing the total weight of picked fruit by the total number of fruits.

## 3. Results

### 3.1. Microclimate Conditions inside and outside the Greenhouse

Table 1 summarizes the recorded climatic conditions, including air temperature (°C), relative humidity (%), total solar radiation (kJ·m^−2^), and photosynthetically active radiation (PAR, mol·m^−2^·day^−1^). The two periods share similar behavior in terms of air temperature and relative humidity values. Temperatures were higher during the day, and relative humidity values were lower at daylight hours. The average values of PAR and SR during period 2 were 75% and 55% higher than those during period 1, respectively.

The lowest values of SR (kJ·m^−2^) and PAR (mol·m^−2^·week^−1^) were obtained under the black net (Figure 4). The beige net showed the greatest PAR values during period 1 compared to other nets, while the green net showed the greatest PAR values during period 2. The average values of transmissivity under control treatment for SR and PAR were 51% and 43%, respectively, during period 1 and 50% and 61%, respectively, during period 2 (Table 2). Among the other treatments, the green net showed the highest transmissivity values for SR (38% and 35%) during periods 1 and 2, respectively, and the highest PAR transmissivity of 34% was achieved during period 2. The beige net showed the highest transmissivity (27%) for PAR during period 1. Jeong et al. (2009) reported similar results. They cultivated begonias (*Begonia cucullata*) in a greenhouse in Columbus (OH, USA) and obtained around 38% SR transmissivity for screens of 50% shading. Lugassi-Ben-Hamo et al. (2010) evaluated the effect of clear plastic screens installed inside a greenhouse on lisianthus (*Eustoma grandiflorum*) and found 12–33% of solar radiation transmissivity. The black net showed the smallest SR transmissivity, followed by the beige net (Table 2). The smaller PAR transmissivity observed in the black net, green net, and beige net treatments in period 1 was also obtained by [24], which indicates that the darker the color of the nets, the greater the capacity of the nets to absorb PAR.

In Figure 5, the ratio of photosynthetically active radiation (PAR) to solar radiation (SR) (PAR/SR) is shown for each treatment and the outside condition. Outside the greenhouse, the PAR/SR ratio was approximately 41%. In the control group, which was affected only by the polycarbonate cover, the PAR/SR ratio increased to 44%. Among the treatments with net coverings, the black net had the highest ratio (39%). The black net allowed the least amount of PAR and SR to pass through, with total SR and PAR values of 746.8 and 293.7 (MJ·m^−2^), respectively. In comparison, the green net had SR and PAR values of 969.6 and 337.2 (MJ·m^−2^), respectively, while the beige net had SR and PAR values of 906.7 and 320.8 (MJ·m^−2^), respectively. According to Shahak et al. [7], the black shading net reduces light transmission to the plants while maintaining its quality. Additionally, Al-Helal and Abdel-Ghany [24] observed that shading nets with shinier colors had higher levels of light transmission, reflecting almost all incident PAR, compared to dark screens that only reflect incident PAR in the specific color band and absorb incident PAR from the remaining complementary colors of the spectrum. Kittas et al. [27] reported that black nets neutralize the PAR/SR ratio and reduce PAR inside the greenhouse. The green and beige nets had the lowest PAR/SR ratio (35%) since these colors promote greater reflection in the wavelengths of the visible light spectrum (PAR). According to Al-Helal and Abdel-Ghany [24], the green and beige nets transmit less PAR than the black net but promote greater PAR reflection.

The daily evaluations during period 1 revealed that the air temperature inside the greenhouse was consistently higher than the outside temperature, with an average difference of at least 2.8 °C (Figure 6). This temperature difference can be attributed to the greenhouse acting as a solar collector, which leads to an increase in temperature. Buriol et al. [28] indicated that the change in air temperature inside greenhouses is influenced by factors such as the size of the greenhouse, ventilation, and transmitted solar radiation. Lugassi-Ben-Hamo et al. [29] found temperature differences of 5.7 ± 2.5 °C when studying different shading nets inside the greenhouse, with shading levels ranging from 67% to 88%.

During period 2, the greenhouse’s inside air temperature was consistently lower than outside, with an average difference of 9.6 °C because of the effectiveness of the shading nets in reducing temperature and the influence of fan-pad cooling. Al-Helal [30] found temperature differences of 0.5 to 9.0 °C due to fan-pad cooling. Among the treatments, the control group had the smallest temperature difference, indicating that the air temperature without shade was higher than in the other treatments, highlighting the impact of the nets in reducing sensible heat. The greenhouse cover’s role in altering solar radiation properties allows for temperature control, as observed by Guiselini [31] with 6 °C temperature differences between the greenhouse covered with white plastic and the thermo reflective screen. However, even with a black and white plastic net combination, the temperature was still approximately 3 °C higher than outside, similar to the findings in period 1 of our study.

Although the colored shading nets had no direct impact on air temperature, they showed a slight effect when a fan-pad cooling system was used during the daytime in period 2 (Figure 6). Specifically, the air temperature values under the black and green nets were lower than those under the beige net. Furthermore, the shading nets influenced the relative humidity values during period 2. The relative humidity values under the shading nets were higher compared to the control condition (without a net) (Figure 6). Among the shading nets, the black net resulted in the highest daytime relative humidity values during period 2. This finding is consistent with the study by Ahemd et al. (2016), which reported that combining shading with ventilation and/or evaporative cooling led to an increase in relative humidity of up to 20%.

Table 3 shows the differences in air relative humidity (ΔRH), saturation water vapor pressure (Δe_s_), and actual water vapor pressure (Δe_a_) between the inside of each shading net and the outside environment during the two periods. In period 1, the variation of relative humidity inside the greenhouse was influenced by temperature and ventilation rate. As the temperature increased, the relative humidity tended to decrease, as reported by Buriol et al. (2000) and Rocha (2002). Since the temperature inside the greenhouse was higher than outside, the relative humidity inside the greenhouse generally remained lower. However, in period 2, the main influencing factor on relative humidity was the fan-pad cooling system rather than temperature and ventilation rate. On average, the black net showed the highest difference in relative humidity compared to the other micro-environments. The remaining treatments exhibited similar relative humidity values among themselves. When evaluating actual water vapor pressure (e_a_), the differences were negative, indicating that there was a higher absolute amount of water vapor inside the greenhouse compared to the outside environment.

### 3.2. Spectral Distribution of Light

The distribution of photon flux under the influence of colored shade nets throughout the experimental period can be seen in Figure 7. The green and beige nets have demonstrated the ability to modify the spectrum or quality of light within the greenhouse.

The green net allowed for greater transmission of light in the blue–green region, specifically within the wavelength range of 400 to 570 nm. Moreover, it transmitted the highest photon flux from 480 nm. On the other hand, the beige net increased the transmission of infrared radiation flux from 730 nm and above. It notably transmitted the highest photon flux from 573 nm onwards.

According to Table 4, the green net exhibited the highest PAR, green light, and blue light, surpassing the transmission rates of the black net by 18.1%, 21%, and 24%, respectively. On the other hand, the beige net demonstrated the highest transmission rates for red and far-red light, exceeding those of the black net by 20% and 25.8%, respectively. These findings highlight the black net’s ability to block incident radiation while showcasing the photo-selective transmission properties of the colored nets. Similar changes in light quantity and quality due to net color have been observed in previous studies conducted by Oren-Shamir et al. [1], Shahak et al. [32], Ayala-Tafoya et al. [33], and Tafoya et al. [34]. These studies evaluated the impact of net color on the cultivation of ornamental plants such as kohuhu (Pittosporum *Variegatum*) as well as various fruits including apple (*Malus domestica*), cherry (*Prunus persica*), grape (*Vitis vinifera*), Japanese persimmon tree (*Diospyros kaki*), pear (*Pyrus communis*), and strawberry (*Fragaria ananassa*).

The green nets increased the (BL/RD) ratio, which exceeded the (BL/RD) ratio of the black nets by 10%. Conversely, the beige nets reduced this ratio by 20% compared to the black nets. Table 5 clearly demonstrates a significant difference in the BL/RD ratios between the green and beige nets, which were measured at 1.067 and 0.771, respectively. When compared to the control, the black net has no effect on the quality of light. Furthermore, both the green and beige nets reduced the red to far-red light ratio (RD/FR) by 4.6% and 3.9%, respectively, in comparison to the RD/FR ratio of the black net, as indicated in Table 5. These findings highlight the ability of the green and beige nets to alter light quality by selectively transmitting certain wavelengths while absorbing or reflecting others. Similar changes in light quality due to net color have been observed in studies conducted by Oren-Shamir et al. [1], Shahak et al. [32], Ayala-Tafoya et al. [33], and Tafoya et al. [34] during their evaluation of tomato (*Solanum lycopersicum*) and cucumber (*Cucumis sativus* L.) greenhouse cultivation.

### 3.3. Physiological Characteristics

The data in Table 6 indicate that strawberries grown under beige shade nets exhibited significantly higher net photosynthetic rates (µmol CO_2_·m^−2^·s^−1^), conductance to water vapor (mol H_2_O/m^2^·s), and transpiration rates (µmol H_2_O·m^−2^·s^−1^) compared to those grown under black shade nets, with increases of 2%, 19%, and 23%, respectively. However, the intercellular CO_2_ concentration for strawberries under the beige shade net was the lowest (121 µmol CO_2_/mol) compared to all other plants. The intercellular CO_2_ concentration was significantly lower (25%) under the beige shade net compared to the concentration observed under the black shade net (161 µmol CO_2_·mol^−1^). This can be attributed to the fact that the beige shade net allowed for the transmission of a larger amount of red light. These findings align with a study conducted by Yanagi et al. [35], which investigated the effect of red light on the photosynthetic rate and transpiration of strawberries. They used different levels of red light intensity and observed that increased red light levels led to higher photosynthetic rates and transpiration rates.

### 3.4. Plant Growth Indices

The growth of strawberry plants was significantly influenced by the environment created by the shade nets, as indicated by Table 7 and Table 8. The leaf traits, including the number of leaves, fresh weight (FW) and dry weight (DW) of leaves, and leaf area, as well as the stem traits, such as plant height, number of crowns, crown length, and fresh and dry weight of crowns, were all higher when the plants were grown under beige shade nets. This can be attributed to the fact that the beige shade net enables an interaction of higher photosynthetically active radiation and red light transmission. In terms of these traits, the green shade net and control treatment ranked second as they allowed for the second-largest transmission of red and blue light. Therefore, the photosynthesis rate increased the biomass production, which generally indicates a greater area of phloem [36,37], a more efficient transport and greater reserve capacity of assimilates for fruit growth [38]. The conditions that allow for environmental enrichment of the environment with diffuse light, spectrally modified by the beige and green net, are photosynthetically more efficient than direct light due to its greater capacity to penetrate the vegetative canopy [1,39]. Colored shading nets can increase light scattering by 50% or more [7,32].

### 3.5. Plant Yield

Yield traits (number of fruits per plant, average fruit weight, and dimensions) were highly influenced by the effect of the light environment created by plastic nets and colored nets on plant physiology (Table 9). The beige net and control treatments exhibited significantly the highest average weight of strawberry fruits, which can be attributed to their positive impact on increasing plant biomass (including both fruits and vegetative parts), enhancing the availability of solar radiation, and improving photosynthetic efficiency. These factors collectively contributed to an accelerated fruit growth rate and a shorter period between anthesis and harvest. This response, along with an increase in fruit number per plant, generally results in increased productivity [40]. Furthermore, Marcelis et al. [41] reported that larger leaf areas allow for more photo-assimilates, which enhance fruit growth and/or support a larger number of fruits. This is often associated with a decrease in fruit abortion rates in plants.

The highest total yield was observed under the beige net, reaching 6.49 kg·m^−2^, while the lowest yield was recorded under the black net at 5.14 kg·m^2^ (Table 10). The yield of strawberries grown under the beige net and control were significantly higher by 26.3% and 13.6%, respectively, compared to those obtained under the black net. No significant differences in yield were observed between plants grown under green nets and other treatments, as well as between plants grown under beige nets and control. It is evident that the black net treatment exhibited the lowest yield among all the treatments. These findings are consistent with those of Ayala-Tafoya et al. [33], who conducted a study in tomato cultivation within a greenhouse and found that the use of black nets with 50% shade resulted in the lowest yields, while a 30% shade pearl net promoted the highest yields. Similar results were reported by Ilić and Fallik [39] and Shahak et al. [42], who utilized red and pearl nets with shade levels ranging from 30% to 40% and reported sweet pepper yields that were 62% to 135% higher compared to those obtained using black nets with the same shade level. These studies collectively support the notion that the choice of shade net can significantly impact crop yield, with certain colored nets proving more beneficial than others in terms of optimizing productivity.

## 4. Conclusions

This study examined the impact of different colors of shading nets on the environmental conditions, spectral light distribution, as well as the growth and yield of greenhouse-grown strawberry plants. Data were collected during two distinct periods: period 1, which represented the cold winter months of December and January, and period 2, encompassing the hot spring months of March and April. The color of the shading nets appeared to have a modest impact on air temperature and relative humidity under the nets during the test period. It was found that green and beige nets had higher transmissivity of photosynthetically active radiation (PAR) and solar radiation (SR) compared to black nets by 5 and 8%, respectively. The color of the nets significantly altered the light spectrum, with green nets transmitting more light in the blue–green region and beige nets increasing infrared radiation flux and transmitting more red light. Green and beige nets had the ability to modify light quality by affecting the ratios of different light wavelengths. They reduced the red to far-red light ratio (RD/FR), and the green nets increased the (BL/RD) ratio, while the beige nets reduced this ratio, while black nets had no effect on the quality of light. The light environment created by the nets had a significant impact on the growth of strawberry plants, with higher leaf and stem traits observed under beige nets. Moreover, the use of beige and green nets resulted in increased average fruit weight and yield, attributed to enhanced plant biomass and photosynthetic efficiency. Notably, the green and beige nets allowed for the highest levels of PAR to reach the plants, at 295.9 and 277.4 μmol·m^−2^·s^−^¹, respectively. Meanwhile, the black net significantly reduced the PAR level to 250.5 μmol·m^−2^·s^−^¹. The highest yields were obtained under beige nets and control treatment. Overall, the study highlights the importance of shading net color in optimizing environmental conditions and maximizing strawberry production in greenhouse settings under arid conditions. Comprehensive studies are needed, with each shading net color treatment applied within a single greenhouse to evaluate the impacts on the overall growing environment and strawberry production.

## Figures and Tables

**Figure 1 plants-13-02318-f001:**
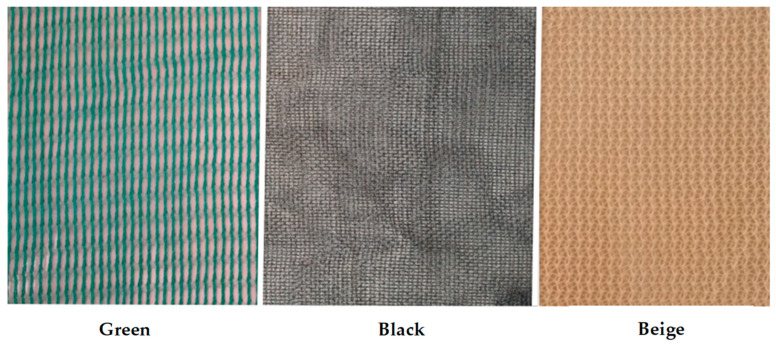
The shading plastic nets used in the study were locally produced. They were rated at 50% shading and were provided by the Saudi Yarn and Knitted Technology Factory (SYNTECH-ISO 9001).

**Figure 2 plants-13-02318-f002:**
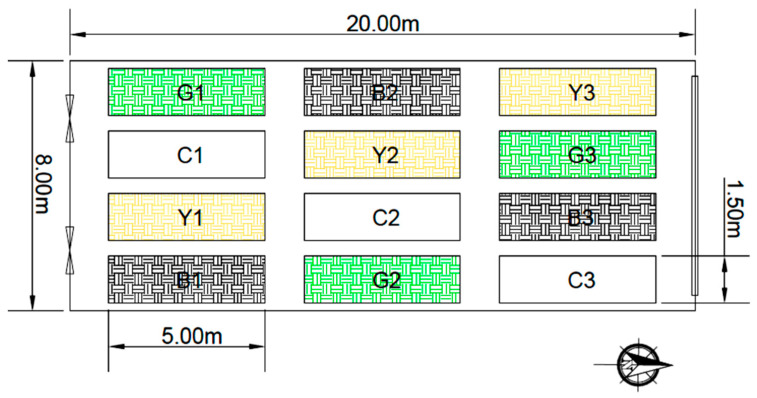
The distribution of the colored shading nets blocks, blocks with black shading nets (B), blocks with green shading nets (G), blocks with beige shading nets (Y), and the control blocks (C). Each shading net covered three blocks (replication) with dimensions of 5 m × 1.5 m × 3.5 m.

**Figure 3 plants-13-02318-f003:**
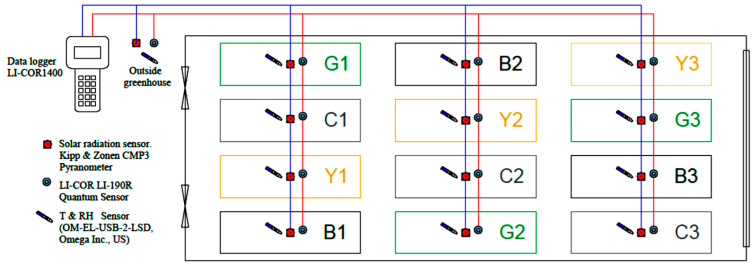
Schematic diagrams explain sensors of the solar radiation, temperature, and relative humidity distribution and types in the greenhouse under shading net blocks and unshading blocks.

**Figure 4 plants-13-02318-f004:**
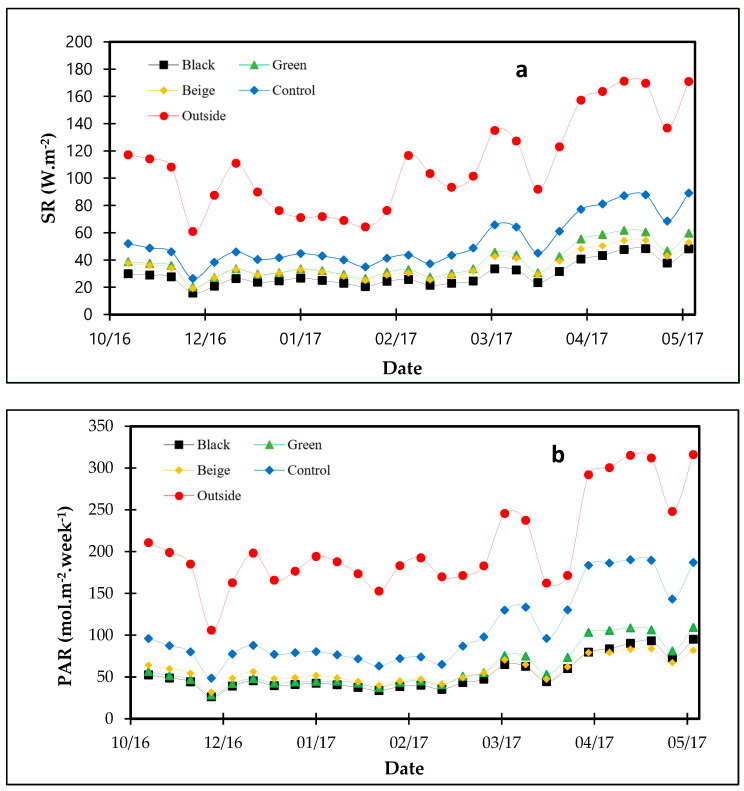
The weekly total of (**a**) solar radiation (SR, kJ·m^−2^) and (**b**) photosynthetically active radiation (PAR, mol·m^−2^·week^−1^) under the influence of colored shade nets (green, black, and beige) and the control treatment.

**Figure 5 plants-13-02318-f005:**
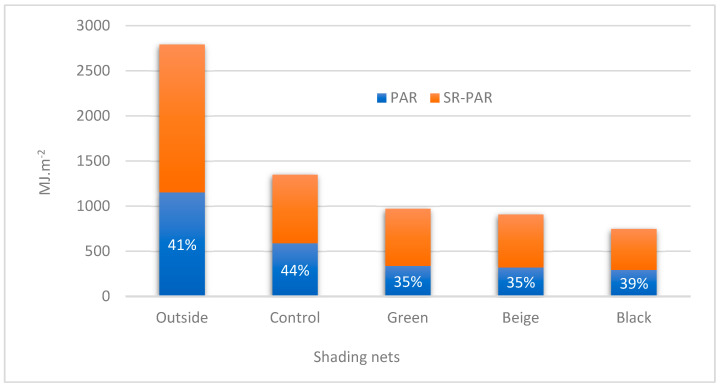
Values of total solar radiation and percentage of photosynthetically active radiation (PAR) in relation to global solar radiation (SR) in all the treatments inside and outside the greenhouse.

**Figure 6 plants-13-02318-f006:**
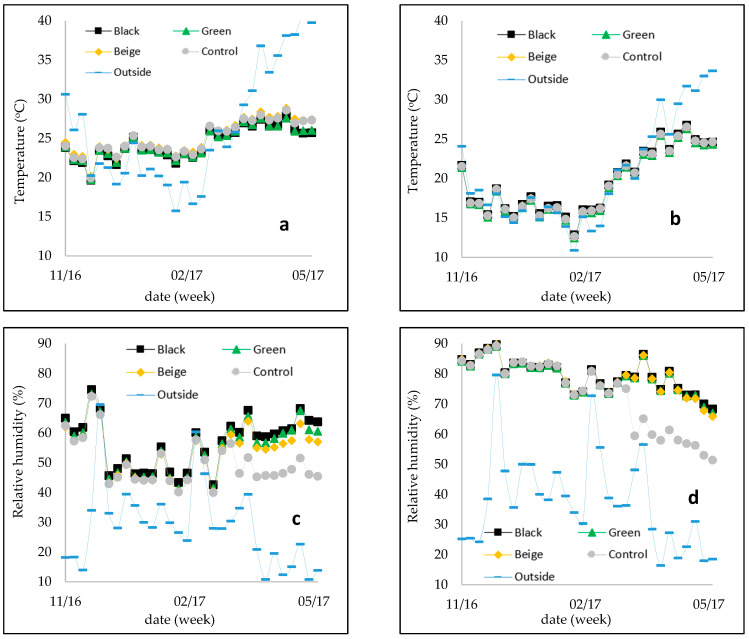
The weekly averages of (**a**) air temperature at daytime (°C), (**b**) air temperature at night time (°C), (**c**) air relative humidity at daytime (%), and (**d**) air relative humidity at night time (%) during the period from November 2022 to May 2023.

**Figure 7 plants-13-02318-f007:**
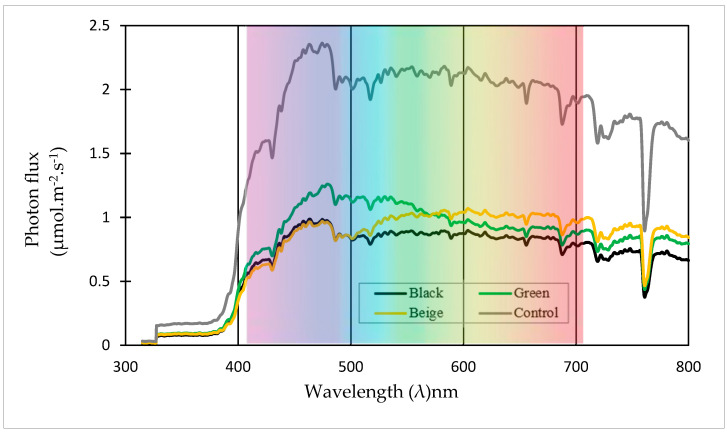
Distribution of photon flux measured between 11:00 and 13:00 h under the influence of colored shade nets and control treatment.

**Table 1 plants-13-02318-t001:** Recorded averages of climatic inside and outside conditions, including air temperature (°C), relative humidity (%), total solar radiation (SR), and photosynthetically active radiation (PAR).

Period	Temperature (°C)				Relative Humidity (%)				Radiation (SR in kJ·m^−2^ and PAR in mol·m^−2^·day^−1^)			
	Inside		Outside		Inside		Outside		Inside		Outside	
	Day	Night	Day	Night	Day	Night	Day	Night	SR	PAR	SR	PAR
1	24	16	21	16	47	82	34	45	4.5	7.3	11.6	25.6
2	27	25	37	30	57	70	16	23	7.2	13.7	20.3	36.1

**Table 2 plants-13-02318-t002:** Percentages of incoming solar radiation (SR) and photosynthetically active radiation (PAR) transmitted into each treatment during the two experimental periods: period 1, representing winter months (December and January), and period 2, which represents spring months (March and April).

Variable	Light Transmission (%)							
	Period 1				Period 2			
	Control	Green	Beige	Black	Control	Green	Beige	Black
SR	51	38	37	30	50	35	32	27
PAR	43	24	27	22	61	34	30	29

**Table 3 plants-13-02318-t003:** Differences of relative humidity (ΔRH), actual vapor pressure (Δe_a_), and saturation vapor pressure (Δe_s_) among the treatments and outside for the two experimental periods: period 1, representing winter months (December and January), and period 2, which represents spring months (March and April).

Variable	Treatment							
	Period 1				Period 2			
	Control	Green	Beige	Black	Control	Green	Beige	Black
ΔRH (%)	12	13	13	15	31	44	41	46
Δe_a_ (kPa)	0.46	0.39	0.47	0.37	−2.55	−2.75	−2.50	−2.75
Δe_s_ (kPa)	0.52	0.52	0.53	0.55	0.77	1.15	1.18	1.21

**Table 4 plants-13-02318-t004:** PAR, blue, green, red, and far-red light (µmol·m^−2^·s^−1^) transmitted by colored shading nets and averaged for the experimental period.

Shading Net	PAR (400–700 nm)	Blue Light (400–500 nm)	Green Light (500–600 nm)	Red Light (600–700 nm)	Far-Red Light (700–800 nm)
Beige	277.4 a,b *	77.6 b	98.3 a	100.5 a	87.6 a
Green	295.9 a	96.9 a	107.2 a	90.8 a,b	79.7 a
Black	250.5 b	80.1 b	86.4 b	83.2 b	69.7 b

PAR = photosynthetically active radiation. * Means with different letters in each column are statistically different (Duncan, *p* ≤ 0.05).

**Table 5 plants-13-02318-t005:** The average values for the blue light to red light ratio (BL/RD) and the ratio of red light to far-red light (RD/FR) measured and calculated during the experimental period.

Shade Net	BL/RD	RD/FR
black (B)	0.963 b *	1.194 a
green (G)	1.067 a	1.139 b
control (C)	0.953 b	1.203 a
beige (Y)	0.771 c	1.147 b

* Means with different letters in each column are statistically different (Duncan, *p* ≤ 0.05).

**Table 6 plants-13-02318-t006:** Influence of shade nets on the photosynthetic rate (P_n_), conductance to water vapor (G_tw_), intercellular CO_2_ concentration (C_i_), transpiration (E), and leaf temperature (T_L_) *.

Treatment	P_n_	G_tw_	C_i_	E	T_L_
	µmol CO_2_·m^−2^·s^−1^	mol H_2_O·m^−2^·s^−1^	µmol CO_2_·mol^−1^	µmol H_2_O·m^−2^·s^−1^	°C
Beige	11.83 a	0.19 a	121 b	3.23 a	24.5 a
Control	11.71 a,b	0.18 a,b	141 a,b	2.78 b	23.7 d
Green	11.69 a,b	0.17 b	154 a,b	2.75 b	24.2 b
Black	11.58 b	0.16 b	161 a	2.62 b	23.9 c

* Means with different letters in each column are statistically different (Duncan, *p* ≤ 0.05).

**Table 7 plants-13-02318-t007:** Influence of shade nets on leaf traits of strawberry plants *.

Treatment	Number of Leaves	Leaf FW (g)	Leaf DW (g)	Leaf Area (cm^2^)
Black	13.8 b *	33.22 c	10.00 c	314.89 c
Green	17.7 a,b	42.35 b	12.53 b	506.1 b
Control	20.5 a,b	44.38 b	12.20 b,c	474.44 b
Beige	22.3 a,b	52.55 a	17.75 a	711.16 a

* Means with different letters in each column are statistically different (Duncan, *p* ≤ 0.05).

**Table 8 plants-13-02318-t008:** Influence of shade nets on stem traits of strawberry plants *.

Treatment	Plant High (cm)	Number of Crowns	Crown Length (cm)	Crowns FW (g)	Crowns DW (g)
black	25.7 a	2.17 c	2.15 a	9.23 b	5.8 a
green	28.2 a	3.43 b	2.4 a	13.53 a	6.8 a
control	26.2 a	3.5 b	2.67 a	14.25 a	6.3 a
beige	29 a	5 a	2.75 a	16.6 a	6.8 a

* Means with different letters in each column are statistically different (Duncan, *p* ≤ 0.05).

**Table 9 plants-13-02318-t009:** Influence of shade nets on fruit traits of strawberry plants *.

Treatment	Number of Fruit	Fruits FW (g)	Fruits DW (g)	Fruit Length (mm)	Fruit Diameter (mm)
black	42.7 a	15.10 c	2.14 b	45.67 a	35.00 b
green	43.0 a	16.97 b	2.41 a	45.11 a	34.89 b
control	43.1 a	18.13 a,b	2.62 a	46.11 a	38.22 a
beige	44.0 a	18.43 a	2.63 a	46.50 a	38.17 a

* Means with different letters in each column are statistically different (Duncan, *p* ≤ 0.05).

**Table 10 plants-13-02318-t010:** Influence of shade nets on the total yield of strawberry plants *.

Treatment	Yield	
	(g per Plant)	(kg·m^−2^)
black	642.9 b	5.14 b
green	729.6 a,b	5.84 a,b
control	780.3 a	6.24 a
beige	810.6 a	6.49 a

* Means with different letters in each column are statistically different (Duncan, *p* ≤ 0.05).

## Data Availability

Data is contained within the article.

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
