# Peer review of "Effects of Shading Nets Color on the Internal Environmental Conditions, Light Spectral Distribution, and Strawberry Growth and Yield in Greenhouses"

_plants, 2024, doi:10.3390/plants13162318_

Round 1
Reviewer 1 Report
Comments and Suggestions for Authors
Comments are in the attached file

Author Response
Response to Reviewer #1 Comments
Reviewer comments are in black lines,
Author responses are in blue lines, and
Modifications through the manuscript are in blue lines.
Thank you for your valuable input. We sincerely appreciate the time, efforts, and valuable comments that greatly enhanced our manuscript. All your suggestions have been considered and incorporated accordingly.
- Major problems:
- A major problem is at materials and methods when describing the plastic shade nets. Technical net characteristics are missing and control treatment characteristics are not even mentioned. Too much text is dedicated to temperature and humidity under the nets. It makes no sense when a cooling system is used. It homogenized the treatments. The effect observed is minor. It would be different if the treatment was applied to all greenhouse?
The available information of the technical details describing shade nets are provided in lines 136-139 as made by the manufacturing company. Also, we measured the spectral distribution of light, which represents the radiative characteristics of each shade net, throughout the growth period using a spectrometer. The results of these light measurements are presented in Table 4 and Figure 7 of the study.
The experiment was conducted in a single greenhouse divided into 12 blocks, using a randomized block design. Using a single greenhouse was necessary, as it would have been difficult to accommodate multiple greenhouses, with each one dedicated to a specific treatment. The study was carried out over two distinct periods; winter months when there was no need for cooling, and spring months when a cooling system was utilized. This reflects the practical application of greenhouse practices in arid regions.
We have added to the conclusion that comprehensive studies are needed, with each shading net color treatment applied within a single greenhouse to evaluate the impacts on the overall growing environment and strawberry production.
- “Moreover, the use of beige and green nets resulted in increased 490 average fruit weight and yield” - This sentence from the conclusions is not supported by paper data.
This sentence from the conclusion is supported by data (lines 472-474).
- Some minor problems:
- Four keywords are in the paper title, avoid it.
This was considered.
- Species names in the beginning of the paper are not in italic.
This was considered and adjusted carefully throughout the manuscript.
- It is well described in the literature that black nets just block’s light. Color nets have morphological effects on plants. This was not well explored by authors.
The results section considers the morphological changes in leaf traits, stem traits, and fruit traits in response to the different colored shade nets (lines 400-452).
- Figure 1 takes too much space and is not needed if explained in the text. The same for figure 4. Figure 8 is not needed. Not significant for the objectives of the study. Figure 11 and 12 are also not needed.
Thank you for the comment. Figures 1, 4, 8, 11 and 12 were omitted from the manuscript.
Response to Reviewer #2 Comments
Reviewer comments are in black,
Author responses are in normal and blue lines, and
Modifications through the manuscript are in blue lines.
Thank you for your valuable input. We sincerely appreciate the time, efforts, and valuable comments that greatly enhanced our manuscript. All your suggestions have been considered and incorporated accordingly.
Manuscript is in fairly good shape. I have a few comments for consideration.
- Capitalize "yield" in the title.
The word “yield” was capitalized in the title.
- SR is used in the abstract but not defined. Please include "solar radiation" with (SR) after the parentheses the first time this is used in the abstract and in the text.
Ok, this was considered, line 23.
- Only have to use "(Fragaria x ananassa) cv. Fortuna" once. After that just use "strawberries." Please include the daylength sensitivity of this cultivar. Is it day-neutral or short-day?
This was considered in the manuscript. The cultivar is a short-day strawberry variety (lines 154 and 155)
- Avoid saying "Figure 6 presents" (line 252) or "Table 2 presents" (line 255) or "Figure 9 shows" (line 325). Just make the statement and cite the table or figure.
This approach was applied consistently throughout the manuscript.
- Line 159 - From where was the plant material obtained?
The source of the plants was added (lines 153 and 154).
- Typically, SI units are expressed with negative superscripts as opposed to, say, kg/ha. This should be written as kg . ha-1.
The SI units throughout the manuscript were changed to be expressed in negative superscripts instead of the regular unit format.
- Typically, figure and table legends require more information. They should stand alone to the reader so one doesn't have to look through the text to find information about the table or figure.
The legends for the figures and tables were revised to include more detailed information.
- In Figs. 6 and 10, "beige" is misspelled in the caption. Also, "Control" is capitalized but the colors are not. Need to be consistent.
Please note that according to reviewer # 1 comments, we omitted Figures 1, 4, 8, 11 and 12 from the manuscript. So, Figure 6 becomes 4 and Figure 10 becomes 7. The captions for Figures 4 and 7 have been updated. The misspelling of "beige" has been corrected, and the formatting has been made consistent - with "Control" capitalized and the color names in lowercase.
- Figs 7 and 8 and 11 - please use regular bar graphs.
Please note that according to reviewer # 1 comments, we omitted Figures 8 and 11 from the manuscript. Figures 7 becomes 5. This Figure has been updated to use regular bar graph formats, instead of the previous presentation.
- Table 9 - Needs more explanation in the legend. Are these numbers individual fruit weight? If so, please define. Observation: I would think that the dry weights should be much smaller than the fresh weights since strawberries are about 85% water.
Thank you for the reviewer comment. The dry weights of the treatments were corrected from the values mistakenly printed in the original manuscript. The correct dry weight values are as follows: 2.14, 2.41, 2.62, and 2.63 g. These revised dry weight measurements have been verified and incorporated into the final data analysis and results section.
- There is a problem with significant figures at various places in the document. Often, too many are used, implying a greater level of precision than actually exists. One example is Table 7. The variable "Number of leaves" has the correct number of significant figures, but was weight measured precisely to the 1/10th of a gram? Was leaf area measured with precision to the nearest square millimeter (six significant figures?)? In Table 9 was fruit diameter measured to the nearest 0.0001 meter?
The figures shown in the tables represent the mean (average) values calculated from the sample data. Therefore, after averaging the individual sample values, the resulting figures are presented with the specified precision, for example, to the nearest one-tenth of a gram or one-tenth of a millimeter.
- Figure 11 and Table 10 show the same data. Just use one of these. I'd prefer the table.
Ok, Figure 11 was omitted and table 10 was used.
We agree that the information in Figure 11 is redundant with Table 10, so we have omitted the figure and used the table.
- Line 490-491 - This is misleading. The green nets had larger yields/fruit higher compared to black, but not compared to the control.
This was considered, line 436-439.
Also, the beige and green nets were compared to the more commonly used black shading nets.
- In the conclusion you should say that the colored nets didn't reduce temperatures relative to the control.
The color of the shading nets appeared to have a modest impact on air temperature and relative humidity under the nets during the test period. This statement was added to the conclusion (lines 460-462).
- There are inconsistencies in the references that should be addressed.
We carefully reviewed the reference list, address any inconsistencies.
Reviewer 2 Report
Comments and Suggestions for Authors
Manuscript is in fairly good shape. I have a few comments for consideration.
Capitalize "yield" in the title.
SR is used in the abstract but not defined. Please include "solar radiation" with (SR) after the parentheses the first time this is used in the abstract and in the text.
Only have to use "(Fragaria x ananassa) cv. Fortuna" once. After that just use "strawberries." Please include the daylength sensitivity of this cultivar. Is it day-neutral or short-day?
Avoid saying "Figure 6 presents" (line 252) or "Table 2 presents" (line 255) or "Figure 9 shows" (line 325). Just make the statement and cite the table or figure.
Line 159 - From where was the plant material obtained?
Typically, SI units are expressed with negative superscripts as opposed to, say, kg/ha. This should be written as kg . ha-1.
Typically, figure and table legends require more information. They should stand alone to the reader so one doesn't have to look through the text to find information about the table or figure.
In Figs. 6 and 10, "beige" is misspelled in the caption. Also, "Control" is capitalized but the colors are not. Need to be consistent.
Figs 7 and 8 and 11 - please use regular bar graphs.
Table 9 - Needs more explanation in the legend. Are these numbers individual fruit weight? If so, please define. Observation: I would think that the dry weights should be much smaller than the fresh weights since strawberries are about 85% water.
There is a problem with significant figures at various places in the document. Often, too many are used, implying a greater level of precision than actually exists. One example is Table 7. The variable "Number of leaves" has the correct number of significant figures, but was weight measured precisely to the 1/10th of a gram? Was leaf area measured with precision to the nearest square millimeter (six significant figures?)? In Table 9 was fruit diameter measured to the nearest 0.0001 meter?
Figure 11 and Table 10 show the same data. Just use one of these. I'd prefer the table.
Line 490-491 - This is misleading. The green nets had larger yields/fruit higher compared to black, but not compared to the control.
In the conclusion you should say that the colored nets didn't reduce temperatures relative to the control.
There are inconsistencies in the references that should be addressed.
Author Response
Response to Reviewer #2 Comments
Reviewer comments are in black,
Author responses are in normal and blue lines, and
Modifications through the manuscript are in blue lines.
Thank you for your valuable input. We sincerely appreciate the time, efforts, and valuable comments that greatly enhanced our manuscript. All your suggestions have been considered and incorporated accordingly.
Manuscript is in fairly good shape. I have a few comments for consideration.
- Capitalize "yield" in the title.
The word “yield” was capitalized in the title.
- SR is used in the abstract but not defined. Please include "solar radiation" with (SR) after the parentheses the first time this is used in the abstract and in the text.
Ok, this was considered, line 23.
- Only have to use "(Fragaria x ananassa) cv. Fortuna" once. After that just use "strawberries." Please include the daylength sensitivity of this cultivar. Is it day-neutral or short-day?
This was considered in the manuscript. The cultivar is a short-day strawberry variety (lines 154 and 155)
- Avoid saying "Figure 6 presents" (line 252) or "Table 2 presents" (line 255) or "Figure 9 shows" (line 325). Just make the statement and cite the table or figure.
This approach was applied consistently throughout the manuscript.
- Line 159 - From where was the plant material obtained?
The source of the plants was added (lines 153 and 154).
- Typically, SI units are expressed with negative superscripts as opposed to, say, kg/ha. This should be written as kg . ha-1.
The SI units throughout the manuscript were changed to be expressed in negative superscripts instead of the regular unit format.
- Typically, figure and table legends require more information. They should stand alone to the reader so one doesn't have to look through the text to find information about the table or figure.
The legends for the figures and tables were revised to include more detailed information.
- In Figs. 6 and 10, "beige" is misspelled in the caption. Also, "Control" is capitalized but the colors are not. Need to be consistent.
Please note that according to reviewer # 1 comments, we omitted Figures 1, 4, 8, 11 and 12 from the manuscript. So, Figure 6 becomes 4 and Figure 10 becomes 7. The captions for Figures 4 and 7 have been updated. The misspelling of "beige" has been corrected, and the formatting has been made consistent - with "Control" capitalized and the color names in lowercase.
- Figs 7 and 8 and 11 - please use regular bar graphs.
Please note that according to reviewer # 1 comments, we omitted Figures 8 and 11 from the manuscript. Figures 7 becomes 5. This Figure has been updated to use regular bar graph formats, instead of the previous presentation.
- Table 9 - Needs more explanation in the legend. Are these numbers individual fruit weight? If so, please define. Observation: I would think that the dry weights should be much smaller than the fresh weights since strawberries are about 85% water.
Thank you for the reviewer comment. The dry weights of the treatments were corrected from the values mistakenly printed in the original manuscript. The correct dry weight values are as follows: 2.14, 2.41, 2.62, and 2.63 g. These revised dry weight measurements have been verified and incorporated into the final data analysis and results section.
- There is a problem with significant figures at various places in the document. Often, too many are used, implying a greater level of precision than actually exists. One example is Table 7. The variable "Number of leaves" has the correct number of significant figures, but was weight measured precisely to the 1/10th of a gram? Was leaf area measured with precision to the nearest square millimeter (six significant figures?)? In Table 9 was fruit diameter measured to the nearest 0.0001 meter?
The figures shown in the tables represent the mean (average) values calculated from the sample data. Therefore, after averaging the individual sample values, the resulting figures are presented with the specified precision, for example, to the nearest one-tenth of a gram or one-tenth of a millimeter.
- Figure 11 and Table 10 show the same data. Just use one of these. I'd prefer the table.
Ok, Figure 11 was omitted and table 10 was used.
We agree that the information in Figure 11 is redundant with Table 10, so we have omitted the figure and used the table.
- Line 490-491 - This is misleading. The green nets had larger yields/fruit higher compared to black, but not compared to the control.
This was considered, line 436-439.
Also, the beige and green nets were compared to the more commonly used black shading nets.
- In the conclusion you should say that the colored nets didn't reduce temperatures relative to the control.
The color of the shading nets appeared to have a modest impact on air temperature and relative humidity under the nets during the test period. This statement was added to the conclusion (lines 460-462).
- There are inconsistencies in the references that should be addressed.
We carefully reviewed the reference list, address any inconsistencies.